# A Systematic Review of Scientific Studies on the Effects of Music in People with or at Risk for Autism Spectrum Disorder

**DOI:** 10.3390/ijerph19095150

**Published:** 2022-04-23

**Authors:** Briana Applewhite, Zeynep Cankaya, Annie Heiderscheit, Hubertus Himmerich

**Affiliations:** 1Department of Psychological Medicine, Institute of Psychiatry, Psychology & Neuroscience, King’s College of London, London SE5 8AF, UK; briana.applewhite@kcl.ac.uk; 2Mental Health Studies Program, Institute of Psychiatry, Psychology & Neuroscience, King’s College of London, London WC2R 2LS, UK; zeynep.cankaya@kcl.ac.uk; 3Department of Music Therapy, Augsburg University, Minneapolis, MN 55454, USA; heidersc@augsburg.edu; 4South London and Maudsley NHS Foundation Trust (SLaM), London SE5 8AZ, UK

**Keywords:** music, music therapy, autism spectrum disorder, ASD

## Abstract

The prevalence of autism spectrum disorders (ASD) is globally increasing, and the current available interventions show variable success. Thus, there is a growing interest in additional interventions such as music therapy (MT). Therefore, we aimed to provide a comprehensive and systematic review of music and people with, or at risk of, ASD. We used the Preferred Reporting Items for Systematic Reviews and Meta-Analysis (PRISMA) guidelines and used PubMed, PsycINFO, and Web of Science as databases, with “music”, “music therapy”, “autism spectrum disorder”, and “ASD” as search terms. Among the identified and screened articles, 81 out of 621 qualified as scientific studies involving a total of 43,353 participants. These studies investigated the peculiarities of music perception in people with ASD, as well as the effects of music and MT in this patient group. Most of the music-based interventions were beneficial in improving social, emotional, and behavioural problems. However, the availability of studies utilizing a rigorous randomized controlled trial (RCT) design was scarce. Most of the studies had a small sample size, and the applied therapeutic and scientific research methods were heterogeneous.

## 1. Introduction

### 1.1. Autism Spectrum Disorder: Symptoms and Diagnosis

The article, “Autistic Disturbances of Affective Contact”, written by Leo Kanner in 1943, presented 11 children who showed an affinity for loneliness and obsessive behaviours but displayed an intact intelligence. This clinical phenomenon is known as autism spectrum disorder (ASD) today [1]. ASD refers to complex neurodevelopment conditions characterised by some degree of impairments in behaviour, communication, and social functioning [2]. The overall incidence has shown that ASD frequently arises during childhood and persists in adolescence and adulthood. It typically becomes visible in the first five years of life [3]. Over the past 50 years, epidemiological studies have confirmed that the prevalence of ASD is increasing globally. The current prevalence of ASD reported by the Center for Disease Control is estimated to be about one in 160 children. ASD is more than four times more prevalent among males than females [4].

According to the Diagnostic and Statistical Manual of Mental Disorders (DSM-5), ASD is characterised by a persistent deficit in social communication, restrictive and repetitive patterns of behaviours, social and occupational impairments, and other areas of functioning [2].

While the exact cause for ASD remains undetermined, literature on developmental disorders indicate that brain abnormalities could explain the reason for ASD in structure, function, and genetic susceptibility [5]. The literature further suggests that ASD is genetically determined. Additionally, environmental influences might also be considered as risk factors for developing ASD [6], and there is an interaction effect of genes and environmental components [7].

### 1.2. Treatments for ASD and Music Therapy

There are several treatments available for individuals with ASD, and treatment plans depend on individual needs. Since autism is on a developmental spectrum, early interventions are essential in improving symptoms, learning, and development [8]. The applied behaviour analysis (ABA) is most commonly used in schools and clinics to help children to improve positive behaviours, and this approach includes different training and interventions. There is no specific medication that can alleviate the core symptoms of ASD. Risperidone and aripiprazole are FDA-approved only for managing the irritability associated with the disorder [9]. As there is a significant degree of co-morbidity between autism and attention-deficit-hyperactivity-disorder (ADHD), psychostimulant medications, such as methylphenidate and amphetamines, may improve ADHD symptoms in patients with ASD [6].

The management of ASD is mainly based on psychosocial interventions [10]. However, there are three possible interventions recommended by the National Institute for Health and Care Excellence (NICE): psychosocial interventions for core symptoms, psychosocial interventions focused on life skills, and biomedical interventions [11].

The prognosis for children with ASD has been poor, as only 25 percent show “good” or “fair” outcomes. Similar rates have been demonstrated for adults with ASD. The majority of adults with ASD remain dependent on their families or professional care. They still have difficulties in formal education, maintaining employment, living independently, and sustaining relationships [12]. Therefore, there is a need for therapeutic, improved treatments and extensive research, which should investigate the efficacy of current interventions [13].

There is a growing interest in alternative and non-pharmacological based interventions, using auditory and sensory integration practices, in the treatment of autism. Promising evidence was found for the use of music therapy (MT). Several studies have strengthened the use of this behavioural approach [14].

MT is an approach that offers a clinical and evidence-based intervention within a therapeutic frame to target the physical, emotional, cognitive, and social needs of an individual. It might be organised as self-help, individual, group-based, or peer-mediated therapies (American Music Therapy Association, n.d.). Group-based MT provides an indirect form of communication, which improves engagement in people with ASD. A study by LaGasse [15] investigated this suggestion, by examining children’s social behaviours with ASD, and noticed that they reported significant progress in joint attention, eye contact, and turn-taking. Therefore, group-based MT aims to use music relationships to encourage patients to develop social connections with others [16].

The purpose of MT is to transfer the skills developed in music-based experiences to other areas of life by developing cognitive, motor, emotional, social, sensory, and learning skills. In both receptive and active music making experiences, there is an activation in the superior temporal lobe and inferior frontal areas of the brain involved in cognitive, sensorimotor, and perception-action mediation areas by increasing the synchrony between these cortical areas, which, in turn, promote heightened sensory integration [17]. People who engage with music long-term show changes in volume and density the most within the cortex and the cerebellum [18]. Children with ASD who are engaged in music experiences long-term have shown larger corpus callosum, frontal, temporal, and motor areas [19].

Both active music-making and receptive music engagement have cognitive benefits for children with ASD—mainly sustained attention, memory, and enhanced verbal communication [20]. This was tested in a study, demonstrating an increase in the functional connectivity between the bilateral primary auditory cortex and sub-cortical and motor regions, in children participating in the music-based intervention compared to a non-music intervention [21].

Music might also be used as a tool to enhance the interpretation and communication of emotions [16]. A study by Katagiri (2009) presented four teaching conditions to determine how people learn emotional concepts: non-purposeful teaching, teaching with verbal intrusions, teaching while background music representing the emotion was played, or teaching while, singing [22]. The findings highlighted that all children showed greater emotional understanding when teaching was accompanied by background music representing the emotion, enhancing empathy and social understanding. Additionally, biological studies have reported that, when music is created or listened to in a social context, some neurohormones, such as oxytocin and neuropeptide, are released by the posterior pituitary gland, which promotes “mind-reading” and empathy in people with ASD [23].

Moreover, it has been shown that music helps to alleviate pain, anxiety, agitation, and depression [24]. Pervasive behaviours, such as developing rigid routines, are typical behaviours of individuals with ASD, and changes in these routines might cause significant stress and aggression. It has been demonstrated that participation in MT reduces anxiety and aggression [25].

Researchers have examined the effects of MT on individuals with ASD with regards to behaviour, psychosocial, intellectual, and interpersonal parameters. However, there is a need for a comprehensive review of the published benefits and potential side effects of the use of music in people with ASD [26,27,28]. This comprehensive overview is the aim of our systematic review.

## 2. Materials and Methods

### 2.1. Search Strategy

This systematic review was conducted following the recommendations outlined in the Preferred Reporting Items for Systematic Reviews and Meta-analysis (PRISMA) [29].

We performed the systematic literature search using the medical databases PubMed, PsycINFO, and Web of Science. The search of the literature was conducted from inception until February 2022. We used the following key search terms: autism spectrum disorder/autism/ASD in conjunction with music/music therapy. The specific search algorithm in PubMed was: ((“autism spectrum disorder”[Title/Abstract]) OR (“autism”[Title/Abstract]) OR (“ASD”[Title/ Abstract])) AND ((“music”[Title/Abstract]) OR (“music therapy” [Title/Abstract])). The search terms for PsycINFO via Ovid were ((autism spectrum disorder, or autism, or ASD) and (music, or music therapy)). The search terms for Web of Science were also ((autism spectrum disorder, or autism, or ASD) and (music, or music therapy)).

### 2.2. Inclusion and Exclusion Criteria

Inclusion Criteria:The studies are published in English or have an English-language abstract available; participants of studies who were diagnosed with or had symptoms of ASD; studies must be ASD and music-related.The full text of the article is available.The studies used music in an experimental or observational study designStudies with measurable results, or outcomes were reported.

Exclusion Criteria:
Studies were excluded if they were systematic reviews or meta analysis’.Studies that were not ASD related were excluded.Studies were excluded if there was a non music related interventionArticles were excluded if they were protocols, hospital reports, evaluation papers, editorials, qualitative studies, syntheses, case reports, personal reviews or essays.Studies were excluded if they were not published in English.Feasibility studies were excluded.Studies without a clearly defined control group were excluded.Articles were excluded if there was a mixture of diagnosed groups not including those with ASD.

### 2.3. Study Selection and Data Extraction

We screened articles reporting studies of any design that assessed the use of music or MT with people with ASD. All articles were included if the full text was available. Articles were included if they described the methodology and measurable results or outcomes were reported. We only included studies in which music was a part of an experimental or observational study design. Only studies published in English were included. Articles were excluded if music was not applied, and measurable outcomes or effects of music were not reported. Only original publications were included, and reviews, meta-analyses, case reports, protocols, editorials, syntheses, qualitative studies, evolution papers, personal essays, feasibility reports, phenomenological studies, and hospital reports were excluded, and duplicates were removed. Articles that mainly dealt with the mental health of musicians, music students, and music therapists were excluded. Only human studies were included, and animal studies were removed. All articles where only hospitals or therapy programmes were described were excluded. All articles were screened and categorised as “included”, “excluded”, and “unclear.” The titles and abstracts of all identified articles were screened independently by another two reviewers. In total, three reviewers were involved.

The data from all included studies were extracted into an electronic summary table, as displayed in Table 1, by specific methodological characteristics: author, year of publication, sample and group size, total participants, study design, questionnaires and research methods, main outcomes, and the significance of main outcomes. The data extraction was based on the study results investigating the effect of music or MT on ASD. All articles were then thematically presented, and the findings were accordingly reported in a matrix. All the extracted data described were critically discussed in the results section.

### 2.4. Data Analysis

After the extraction of study details, the articles were thematically arranged based on the study design and types of intervention by both B.A, Z.C., and H.H. The findings were then reported accordingly.

### 2.5. Ethical Considerations

As our systematic review did not require the recruitment of probands or patients and did not use data or specimens of individuals, ethical approval is not required. However, we checked the ethical aspects and approval of all included studies.

## 3. Results

### 3.1. Included Studies

Following three searches using PubMed, PsycINFO, and Web of Science databases, 887 candidate papers were identified, and two studies were added through hand-searching and reference-chaining. After the removal of duplicates, a total of 621 articles were assessed for inclusion. A total of 81 studies met the full eligibility criteria and were chosen for analysis. These studies contained data of *n* = 43,353 patients or study participants. Figure 1 shows a PRISMA diagram describing the results of the search strategy and reasons for exclusion. Table 1 summarises all the publications that met the eligibility criteria and were included in this systematic review.

### 3.2. Methodological aspects

The design of the studies, their main results, and their statistical significance are presented in Table 1. Of the 81 identified studies, 3 were surveys, 45 were experimental studies, and 25 studies were longitudinal studies or randomised controlled studies (RCTs) investigating the effect of music and MT. Most of the studies were relatively small with ~50 participants. However, we identified four meaningful studies with more than 100 study participants each.

The largest survey was performed by Ruan et al. (2018) including 34,749 parents of children around the age of three years [94]. Researchers found that antenatal music training and maternal talk to the foetus were associated with a reduction in autistic-like behaviours in children, with a dose-dependent relationship [94].

A large and well-designed experimental study by Goris et al. (2020) with 161 participants from university students investigated music preference in relation to autistic traits by presenting tone sequences that varied in predictability [37]. They found a positive correlation between autistic traits and a preference for predictability in music [37].

The RCT with the most advanced study design was the TIME-A randomized clinical trial by Bieleninik et al. (2017), which was conducted in nine countries and enrolled N = 364 children, aged four to seven years, with ASD [70]. Patients were either allocated to enhanced standard care (*n* = 182) or to enhanced standard care plus improvisational music therapy (IMT) (*n* = 182). In IMT, trained music therapists sang or played music with each child, attuned and adapted to the child’s focus of attention, to help children develop affect sharing and joint attention. The primary outcome measure was symptom severity over 5 months, based on the Autism Diagnostic Observation Schedule (ADOS). However, the mean ADOS social affect score changes did not differ significantly between groups [70].

### 3.3. Content of Included Studies

In the following paragraphs, we provide a comprehensive summary of the content of the studies that fulfilled the inclusion criteria. Table 1 contains further details about each study. Content-wise, the studies fell into four categories: studies on the specific perception of music and music preferences in people with ASD; studies on the effect of music in people with ASD; studies investigating the effect of MT and musical training in people with ASD and their caregivers; studies reporting combined creative arts therapies, including MT, in patients with ASD.

#### 3.3.1. Music Perception in ASD

People with ASD have been found to use music for various purposes such as to address their cognitive, emotional, and social needs including mood management, personal development, and social inclusion [30]. They were reported to have a strong preference for music over verbal material [31], superior pitch memory, recognition of changes of the pitch [34,42,45,48], and to enjoy dissonant music more than controls with typical development (TD) [53]. It’s been shown that people with ASD were physiologically more responsive to their preferred music than those in the comparison group [42]. In a peer-assisted learning study by Johnson and LaGasse (2021), music appeared to increase pro-social skills in ASD children that were paired with neurotypical children [45]. People with ASD appeared to prefer more predictable [37] and upbeat music [31]. The latter may be related to increased activity in dorsolateral prefrontal regions in response to happy music [36].

In contrast to their ability to understand musical cues, as well as the affective and emotional content of music, comparable to non-autistic people [40,43,50], and unlike their similar magnetic resonance imaging (MRI)-measured activation pattern in cortical and subcortical brain regions after hearing music [33], impairments in prosodic language processing were found [35,41,47,51], suggesting alternate mechanisms of speech and music processing in ASD. MRI studies by Sharda et al. (2015), Lai et al. (2012), and Hesling (2010) indicate that the difficulties in prosodic language processing may be due to reduced integrity of the language-processing brain networks, such as the frontotemporal tract, the inferior frontal, and the supramarginal gyrus [35,41,47,51]. However, people with ASD seem to activate their bilateral temporal brain networks during sung-word perception similar to healthy people [51].

Interestingly, compared to TD children, children with ASD do not have a strong preference for a human therapist or human voice over interacting with a robot while dancing or hearing an artificial voice [31,46].

Whipple et al. (2015) compared recognition of symbolic representations of emotions or movements in music between children with severe to profound hearing loss or ASD and TD children, but no significant difference between the ASD, the TD, or the hearing loss group were found [53].

#### 3.3.2. Effect of Music on People with ASD

Studies on the effect of music in people with ASD found that music increases their emotional recognition and comprehension [22,57,64], including increased sign and spoken word imitation [58], help with process socially significant auditory signals [65], and focusing attention [63]. Music was reported to increase motivation for physical exercise [66]. Additionally, adults with ASD reported poor auditory imagery in comparison to health controls [54].

Heaton (2003) found that children with ASD had enhanced pitch memory and tone labelling [59]. Stephenson et al. (2016) tested the response of children and adolescents with ASD to music-evoked emotions using skin conductance and found that participants with ASD showed a reduction in the skin conductance response [62]. However, there was a significant interaction effect with the age group, and the authors interpreted their findings with caution.

Lundqvist et al. (2009) found that vibroacoustic music reduced challenging behaviours, in children with ASD, as well as reduced the frequency of self-injurious behaviours [60]. A study conducted by Portnova et al. (2018) found that children with ASD perceive most pieces of music similarly to neurotypical children except in one musical fragment, where children with ASD found the musical fragment to be “angry and frightening”, whereas neurotypical children perceived the fragment as “sad” [61].

Boorom et al. (2020) examined preschoolers with ASD and their parent’s responsiveness comparing musical and non-musical engagement in 12 parent-child dyads [56]. However, they did not find a significant difference between parental responsiveness in the playing with musical toys condition compared to the non-musical condition [56]. Bhatara et al. (2009) investigated the impact of musical animation videos on social attribution in adolescents with ASD or TD [66]. In their study, adolescents with ASD perceived and integrated music soundtracks with visual displays similarly to TD individuals [55].

#### 3.3.3. Effects of MT and Musical Training on People with ASD

A large survey study showed that regular antenatal exposure to music and talking to the baby might prevent traits of ASD [94]. In small studies, MT was shown to be beneficial regarding the bond between ASD children and their parents [84], movement coordination [73], social communication, interaction and attention [21,26,28,70,71,75,82,90,92,93,96,98], and their overall ASD symptoms [71,75,92,93]. Listening to musical excerpts increased vocal recognition and ability to not hide emotions [91].

However, these positive effects could not be replicated in a large RCT, with 362 participants testing the specific effects of IMT on ASD [70]. The findings of Bieleninik et al.’s study were reported by Crawford et al. (2017), and they were later used by Mössler et al. (2020) to examine the relationship between attunement and changes in the above-mentioned outcomes. However, such a relationship could not be demonstrated.

A study by Simpson et al. (2013) found that children with ASD were more engaged in the sung intervention compared to the spoken condition [20]. Additionally, the use of infant-directed singing was more engaging for children with ASD over infant-directed speech. However, in a follow-up study, Simpson et al. (2015) found there was no significant difference between the sung and spoken conditions [96], although there was a significant increase in receptive labelling skills after both the sung and spoken conditions were enacted [96].

The level of functioning in individuals with ASD [85] and the length of the music intervention might play a significant role in its effectiveness [87].

Active music-making in music therapy sessions seemed to produce significant improvements in social skills in children with ASD [69]. MT also seems to have an impact on biological parameters of neuroplasticity and stress, such as levels of nerve growth factor (NGF) and salivary α-amylase (sAA) [88,91]. Musical training has been found to have a beneficial impact on sound sensitivity [68], sensory gating, and attention [83].

IMT was shown to produce improvements in communicative behaviours [76], as well as their self-regulation, engagement, behavioural organization, and two-way purposeful communication in children with ASD [72]. IMT has also been shown to increase emotional attunement and behavioural synchrony [99]. However, in a study using Relational Music Therapy (RMT), the effects on communication skills were inconclusive [77].

A study by Kern and Aldridge (2006) found that, when music was utilized on the playground, there was an improvement in peer interaction in children with ASD; however, there was no increase in social interactions [81].

A recent study by Jin et al. (2020) examined the efficacy of acupoint therapy on joint attention and social communication [79]. MT was part of both the control and the observation group. Researchers found that both interventions led to an improvement of autistic symptoms in several domains [79]. However, the specific effect of the MT element could not be determined, as MT was included in both treatment conditions [79].

Mössler et al. (2019) investigated the effect of therapeutic relationships, during MT, on changes in social skills at 5 and 12 months [89]. They detected that the therapeutic relationship might be an essential predictor of the development of social skills, language ability, and social communication [89]. However, Mössler et al. (2019) did focus on the effect of MT on social communication. Lim and Draper (2011) compared music and speech training in children with ASD and found that both were effective but not statistically significantly different [86].

Kalas (2012) ascertained that joint attention responses of children with ASD depended on the complexity of the music and their level of functioning [80].

#### 3.3.4. Effects of combined MT and Dance Movement Therapy

A few studies utilized the combination of MT and dance movement therapy (DMT) with ASD populations. A study by Bergmann et al. (2021) utilized the Autism-Competence-Group (AutCom), which includes a combination of psychoeducation with music and dance movement interventions in adults with ASD [101]. The combined intervention produced significant improvement in social competence compared to the control group and emotional competence in the pre-post self-assessment on the AutCom questionnaire [101].

One study found a reduction in compulsive and stereotyped behaviours in a combined music and dance intervention [102]. Another found that a combination of MT and DMT reduced ASD scores in both the experimental and control group [103].

## 4. Discussion

### 4.1. Overview of the Main Results of This Systematic Review and Their Significance

The main topics covered by the studies that fulfilled the inclusion criteria of this systematic review were the specific perception of music and music preferences in people with ASD, the effect of music in people with ASD, and the specific effect of MT and musical training in people with ASD and their caregivers. Studies revealed that people with ASD use music for various purposes [30].

In the first category of identified studies, which investigated music perception in ASD, we found that people with ASD have a superior ability of pitch memory and recognition compared to normal controls. Individuals with ASD were also reported to have a strong preference for music over verbal material. In contrast to their deficits in interpreting the prosody of spoken word, people with ASD have no problems in recognizing and understanding the emotional, communicative, and social aspects of music [36,40,43,50]. Additionally, they seem to have a superior pitch memory and recognition of pitch changes [34,38].

In the second category of articles which investigated the effects of music, studies reported that listening to music can increase emotional recognition and understanding [22,57,64], focus attention [63], and motivate them to do physical exercise [66].

In the third category of articles, which focused on the effects of MT and musical training on people with ASD, MT was shown to be beneficial for the child-parent relationship [84], movement coordination [73], social communication, interaction, attention [15,21,28,74,78,82,87,97,100], and overall ASD symptoms [71,92,93]. However, these effects of MT could not be replicated in an RCT with 362 participants [70]. Studies focusing on biological outcomes indicated that MT has a beneficial effect on biological parameters of neuroplasticity and stress [88,91]. In contrast to language-processing brain circuits, brain areas, connectivity, and function related to the perception and understanding of music seem to be intact in people with ASD [41,47,51].

Only 10 out of the included 81 studies focused on adults and the effects of music, MT, and creative arts therapies on ASD [40,41,43,47,48,62,66,70,80,88]. However, similar results were observed in comparison to adolescent ASD studies. For example, adults with ASD had a reduction in challenging behaviours associated with ASD [62] and significant improvements in social competence compared to control groups [80]. Thus, there is clearly more research available on music and ASD in children and adolescents than in adults. Future research should, therefore, include more samples of adults or studies involving both children and adults with ASD. This way, comparisons could be made about the effects of music and MT in different age groups.

Studies focusing on a mixture of both MT and DMT constitute the fourth category of identified articles. These studies showed that a combination of MT and DMT produced an increase in social competence, as well as a reduction in compulsive and stereotyped behaviours and an overall reduction in ASD scores. This underscores the potential of not just music therapy but other creative arts therapies being possibly effective therapies for individuals with ASD. However, further exploration should be conducted into the use of other creative arts therapies in those with ASD and their potential combined effects.

### 4.2. Limitations

Although the current review yielded promising results, these should be interpreted with caution and considered with several limitations.

Data from RCTs were limited; 17 RCTs were included, and the sample size of most RCTs was small. Therefore, further accumulation of RCT data is necessary to draw firm conclusions. The majority of all included studies had a small sample size, resulting in greater clinical heterogeneity in patients compared to studies with large sample sizes affecting the outcome of the experimental treatment [104].

The majority of studies included child-dominant samples; few studies investigated adults or adolescents with ASD, and no studies examined the elderly population. Therefore, further studies might target different populations of varied different age groups.

Furthermore, there is a lack of longitudinal studies; only 5 out of 81 articles were longitudinal studies, indicating that only short-term effects of music-based interventions have been investigated. Therefore, long-term effects should be investigated.

There was great heterogeneity of study design, outcomes, and MT approaches. Due to different approaches used, there are no confirmatory studies available for any of the reported positive findings. Therefore, the reproducibility of the results has not been shown yet. Additionally, because of the heterogeneous study design, we didn’t conduct a quality assessment which could have led to the inclusion of low-quality studies assessed in the review.

MT was used as an adjunct therapy in all the studies, and this makes it difficult to ascertain the pure effect of the MT. Furthermore, it is difficult to recommend a specific approach of MT because different approaches, as individual vs. group-based MT or active music making vs. passive MT, were conducted. Therefore, we cannot conclude that a specific approach is most effective. Moreover, studies investigated many different and unrelated symptoms, including social skills, emotional understanding, auditory processing, family well-being, and so forth, making the interpretations of the findings difficult.

Another important aspect of utilizing a music intervention with ASD populations is the music preferences of the study subjects. In one study, ASD participants cited music to be “angry and frightening”, whereas neurotypical children cited the same piece of music as “sad” [61]. This highlights the fact that children with ASD perceive music differently, so in studies using a music intervention, it is important to collect the opinions of music preference and individual music sensitivities within ASD participants before beginning the intervention.

Considering the above findings in the context of the wider literature, further investigation should be conducted in different samples, including adults with ASD and elderly individuals with ASD, utilizing larger sample sizes. Additionally, more RCTs should be conducted in order to see the pure effect of MT on ASD-related symptoms. Moreover, ASD symptoms might be investigated as a comprehensive assessment in order to see the impact of music or MT on the more specific symptoms of ASD.

However, as people with ASD use music in many domains and seem to have specific talents regarding analysing and understanding music, this clinical field seems promising. Some studies explored how patients with ASD related positively to an artificial voice or a robot during a musical activity [31,46]. Their unique technical approachability might be kept in mind when designing individually tailored MT for people with ASD in the future.

## 5. Conclusions

The current systematic review demonstrates that the perception of music in people with ASD differs from the music perception in neurotypical controls. Additionally, it shows that music and MT can be used as a therapeutic tool in the management of ASD.

People with ASD seem to have a superior ability of pitch memory and recognition compared to normal controls.

When MT was utilized as an adjunct for managing ASD or ASD symptoms, recognition and understanding of the emotional, communicative, and social aspects of music was elevated, attention was heightened, and motivation for physical exercising was increased. Moreover, music and MT might prevent ASD traits during pre-birth, support the child-parent relationship, and improve movement coordination, social communication, interaction and attention, and overall ASD symptoms in people with, or at risk for, ASD. Music and MT had a beneficial effect on biological parameters. Similar positive results were reported by studies that tested MT in combination with DMT. However, these findings do need to be interpreted regarding of the methodological issues and the potential for publication bias (Thornton and Lee 2000).

These results should be interpreted with caution since findings could not be replicated in an RCT [70]. Different samples should be investigated in order to see whether music and MT might be a mediator for the relationship between age and ASD symptoms. Moreover, more RCTs should be conducted using MT as a comprehensive assessment in order to understand the impact of music and MT on the more specific symptoms of ASD.

## Figures and Tables

**Figure 1 ijerph-19-05150-f001:**
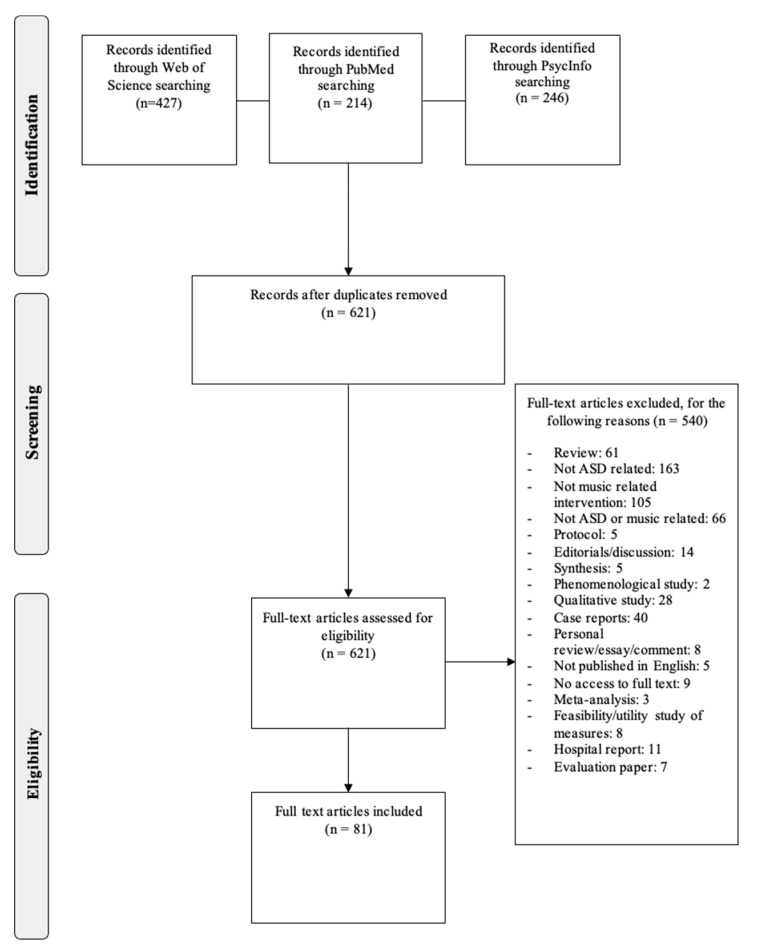
Flow-chart of literature search according to PRISMA.

**Table 1 ijerph-19-05150-t001:** Summary of the studies included in the systematic review. Studies are listed in alphabetical order.

No.	Author (Year), Country	Sample Size and Group Size (*n*)	Total *N*	Study Design	Questionnaires and Research Methods	Main Outcomes	Statistical Significance of Main Results
	**(1) Music Perception in ASD**						
1	Allen et al. (2009), United Kingdom [30]	Adults with ASD (*n* = 12)	12	Survey	Semi-structured questionnaire, early musical experience questionnaire	Most participants exploit music for a wide range of purposes in the cognitive, emotional and social inclusion. However, ASD’s group’s descriptions of mood states reflected a greater reliance on internally focused (arousal) rather than externally focused (emotive) language.	None reported.
2	Barnes et al. (2020), USA [31]	Children with ASD (*n* = 3); TD children (*n* = 12)	15	Experimental design	Preliminary interviews, music selections, dance freeze game	Significant differences between TD children and children with ASD in mimicry, dance, quality, and game play. Greater attention and engagement in ASD children while dancing with a robot.	None reported.
3	Blackstock (1978), USA [32]	Children with ASD (*n* = 10); TD children (*n* = 10)	20	Experimental design (including 2 experiments)	Two pre-recorded 45 minutes cassettes	Children with ASD preferred music over verbal material; they listened to both types of material predominantly with the left ear.	Δ preferred musical material in children with ASD: *p* < 0.001
4	Caria et al. (2011), Germany [33]	Adults with ASD (*n* = 8); TD adults (*n* = 14)	22	Experimental design	ADOS-G, GADS, KADI, WAIS-R, 20 musical excerpts, SAM, schematic pictorial representations, fMRI	Compared to TD group, ASD group showed a decreased brain activation in the premotor area and in the left anterior insula during happy music conditions	Decreased activity in responding to happy compared to sad music in ASD group: *p* < 0.01
5	DePape et al. (2012), Canada [34]	Children with ASD (*n* = 27); TD Children with (*n* = 27)	54	Experimental design	Pitch discrimination task, absolute pitch task, harmonic priming task, Digit Span Subtest of the Weschler Memory Scale-III, PPVT-III, McGurk Auditory-Visual Integration Task, phoneme categorisation, metrical categorisation, hearing thresholds, competing sentences test, Lieter International Performance Scale, background information form	TD children showed higher improvement in filtering, audio-visual integration, specialisation for native phonemic and material categories and lower absolute pitch ability compared to ASD group. No significant group differences were found in harmonic priming performance.	Δ absolute pitch in children with ASD: *p* < 0.001
6	DePriest et al. (2017), Germany [35]	Children with ASD (*n* = 33); TD children (*n* = 17)	50	ERP-based experimental design	ADOS, ADI-R, MWB, LPS, Section 4 test for Non-Verbal IQ, Handedness Questionnaire, brief in-house questionnaire, experimental paradigm, EEG recordings	Language CPS test: No difference between TD and ASD children; music CPS: preserved processing of musical cues in individuals with ASD, but prosodic impairment.	n.s
7	Gebauer et al. (2014), Denmark [36]	Children with ASD (*n* = 23); TD children (*n* = 20)	43	Experimental design	WAIS-III, musical background, questionnaire, musical ear test, emotional stimuli, fMRI	Both groups activated similar neural networks during processing of emotional music; children with ASD showed increase activity in response to happy compared to sad music in dorsolateral prefrontal regions in the rolandic operculum/insula which reflects an increase cognitive processing and physiological arousal in response to emotional musical stimuli.	∆ increased activity in responding to happy compared to sad music in ASD group: *p* < 0.01
8	Goris et al. (2020), Belgium [37]	Children with ASD (*n* = 161)	161	Experimental design	AQ, SRS, music preference paradigm, perceptual fluency, paradigm, gambling paradigm	Positive correlation between autistic traits and a preference for predictability in both music preference and perceptual fluency task; no correlation between autistic traits and gambling behaviour.	∆ music preference paradigm: AQ: *p* = 0.05, SRS: *p* < 0.01; ∆ perceptual paradigm: AQ: *p* = 0.03, SRS: *p* = 0.045; ∆ gambling paradigm: AQ: *p* = 0.63, SRS: *p* = 0.40
9	Heaton et al. 2018), United Kingdom [38]	Children with SLI (*n* = 15); children with ASD with co-morbid ALI (*n* = 15); TD children (*n* = 16)	45	Experimental design	BPVS-II, digit-span sub-test from CMS, parental report questionnaire, Raven’s Progressive Matrices, musical imagery task	ALI participants performed as well as TD children on the tempo condition and better than TD children on the pitch condition of the task. Auditory short-term memory and receptive vocabulary impairments similar across ALI and SLI groups; not associated with a deficit in voluntary musical imagery performance in the ALI group.	Δ pitch performance of ASD group: *p* = 0.16 Δ no correlation between voluntary musical imaginary performance and short-term memory and receptive vocabulary impairment in both ALI and SLI groups: *p* = 0.604
10	Heaton et al. (2007), United Kingdom [39]	Children with ASD (*n* = 20)	20	Experimental design	Seven chord sequences followed by a target chord, experimental task to judge.	No group differences in global and local musical contexts that influence participants’congruity judgments.	n.s
11	Heaton et al. (1999), United Kingdom [40]	Children with ASD (*n* = 12); children with Asperger Syndrome (*n* = 2)	14	Experimental design	PPVT, Raven’s Matrices, verbal IQ, two schematic faces, four bar melodies	Groups did not show any significant difference in their ability to describe musical examples.	n.s
12	Hesling et al. (2010), France [41]	Adults with ASD (*n* = 8); TD adults (*n* = 8)	16	Experimental design	PEPS, fMRI	Correlation between perceptive and productive prosodic deficits for prosodic components, including rhythm, emphasis and affect in HFA. Neural network involved in prosodic speech perception exhibits abnormally activation of the left SMG compared to controls, and an absence of deactivation patterns in regions involved in the default mode.	Δ activation left SMG-turn-end task: *p* < 0.01; Δ focus task: *p* < 0.05; Δ chunking task: *p* < 0.05; Δ focus task: *p* < 0.05; Δ affect task: *p* < 0.05
13	Hillier et al. (2016), USA [42]	Males with ASD (*n* = 23); healthy controls (*n* = 22)	45	Experimental design	ASQ, North American Adult Reading Test, STAI, SAM, electrodermal recording, Pachelbel’s Canon in D major	Participants with ASD physiologically more responsive to their preferred music than those in the comparison group. ASD participants did not differ from controls in their responses to a piece Pachelbel’s Canon.	Δ physiological responsiveness: *p* = 0.007
14	Järvinen et al. (2016), USA [43]	Children with ASD (*n* = 17); children with WS (*n* = 12); TD children (*n* = 20)	49	Experimental design	FISH, probes, ADOS, ADI-R, WISC-III, VIQ, PIQ, FSIQ, nonlinguistic vocal sounds (24) from Montreal Affective Voices, musical pieces by Marsha Bauman of Stanford University, EDA and ECG measures	All three groups: similar emotion identification scores in cognitive abilities across. ASD group: lower autonomic reactivity to sad human voice, increased arousal to vocalisation.	Δ total score: *p* = 0.17; Δ social awareness: *p* = 0.012; Δ social motivation: *p* = 0.034; Δ arousal to vocalisation: *p* < 0.05
15	Järvinen & Heaton (2007), USA [44]	Children with ASD or Asperger syndrome (*n* = 19) TD children (*n* = 19)	38	Experimental design	BPVS, RSPM, same or different discrimination paradigm of music and/or speech stimulus pairs	No difference in pitch sensitivity was found across conditions in the autism group, while children with TD exhibited significantly poorer performance in conditions incorporating speech.	Δ poorer performance in incorporating speech: *p* < 0.001
16	Johnson & LaGasse (2021), USA [45]	Children with ASD (*n* = 18); neurotypical children (*n* = 28)	46	Experimental design	SRS, CARS-II, Peer assisted learning, music creative product-making task	Increase in pro-social skills for some children grouped with NT peers; amount of time in creative music-making was similar between NT and NT/ASD peer groups.	Significant group difference ∆ music task alone vs. with peer: *p* = 0.049
17	Kuriki et al. (2016), Japan [46]	Adults with ASD (*n* = 14); TD Adults (*n* = 14)	28	Experimental design	12 songs, humanness evaluation, familiarity rating assessment	Adults with ASD had similar impressions of humanness and positive feelings for the songs sung by the human and artificial voices.	Δ impressions of humanness and positive feelings: *p* = 0.07
18	Lai et al. (2012), USA [47]	Children with ASD (*n* = 36) TD children (*n* = 21)	57	Experimental design	MRI, ADI-R, Language and communication sub-scale of the ADOS-R, clinical observation; music affinity ratings; fMRI	All three groups had similar emotion identification scores in cognitive abilities; ASD group: lower autonomic reactivity to sad human voice and increased arousal to vocalisation.	Δ n.s between-group difference in rating of music affinity; Δ parents’ between-group difference in children’s affinity for familiar songs: *p* = 0.126
19	Masataka (2017), Japan [48]	Children with ASD (*n* = 19) TD Children (*n* = 28)	47	Experimental design (including 3 experiments)	Mozart’s simple minuet, semi-structured parental interview, harmonic, and inharmonic version of a head-turn preference procedure, four musical pieces of Mozart sonatas, ADI-R, spatial tasks, AP	Both groups preferred the original version of the piece over the inharmonic version. Children with ASD tended to show preference for aesthetic quality of the high dissonant music compared to TD children. AP task: Some children with ASD showed extraordinary musical memory.	Δ dwell time: *p* = 0.02 in face/body looking in singing; Δ familiarity in singing: *p* = 0.02; Δ interaction between familiarity and body/face looking *p* = 0.020; Δ familiarity and looking: *p* = 0.005
20	Mottron et al. (2000), USA [49]	Children with ASD (*n* = 13) TD children (*n* = 13)	26	Experimental design	ADI-R, twelve melodies, same/different judgmental task	Children with ASD performed better in the detection of change in non-transposed, contour-preserved melodies compared to TD children.	Δ performance on detection of change in contour-preserved melodies: *p* < 0.01
21	Quintin et al. (2011), Canada [50]	Adults with ASD (*n* = 26); TD adolescents (*n* = 26)	52	Experimental design	WASI, Digit Span and Letter-Number sequencing subtests of the WISC-IV, SAMMI, SRS, SCQ, a musical task	Adolescents with ASD rated the intensity of the emotions similarly to TD adolescents and reported greater confidence in their responses when they had correctly recognised the emotions.	Δ ratings of emotions intensity: *p* = 0.15; Δ confidence of responses: *p* < 0.01
22	Sharda et al. (2015), India [51]	Children with ASD (*n* = 22) TD children (*n* = 22)	44	Experimental design	ADOS-G, CARS-II, WASI, VIQ, VABS, fMRI	Children with ASD activated bilateral temporal brain networks during sung-word perception; ASD: spoken-word perception right-lateralised and reduced IFG activity; diffusion tensor imaging: reduced integrity of the left hemispheric frontotemporal tract in the ASD group.	Δ decreased IGF activity during sung words: *p* = 0.042
23	Stanutz et al. (2014), Canada [52]	Children with ASD (*n* = 25) TD children (*n* = 25)	50	Experimental design	Music Game 1, brief IQ measure of the Lieter-R paired single-tone pitch discrimination task, melodic memory encoding task, Music Game 2	Improved pitch discrimination ability in the single-tone and melodic context, as well as superior memory for melody and a positive correlation between pitch memory and performance on non-verbal build reasoning ability was found for children with ASD.	Δ single-tone: *p* < 0.05; Δ melodic context: *p* = 0.05; Δ superior memory for melody: *p* < 0.05; Δ positive correlation between pitch memory and performance: *p* < 0.05
24	Whipple et al. (2015), USA [53]	Children with ASD (*n* = 35); Children with severe/profound Hearing loss (*n* = 24); TD children (*n* = 35)	85	Experimental design	PEMM, PPVT-III, EVT, CELF-4	No significant difference between ASD and TD-NH groups in identification of musical emotions or movements.	Δ identification of musical emotions: *p* = 0.97; movements: *p* = 0.1352
	**(2) Effects of Music in People with ASD**						
1	Bacon et al. (2020), United Kingdom [54]	Adults with ASD (*n* = 17); healthy adult controls (*n* = 17)	34	Experimental design	AQ, BAIS, BAIS-V, BAIS-C, earworm questionnaire	Poorer auditory imagery reported in the ASD group for all types of auditory imagery; ASD group did not report fewer earworms than matched controls.	∆BAIS-V: *p* = 0.008
2	Bhatara et al. (2009), Canada [55]	Adolescents with ASD (*n* = 33) TD adolescents (*n* = 26)	59	Experimental design	WASI, Digit Span and Letter-Number Sequencing subtests ofthe WISC-IV, SAMMI, SCQ, SRS	Adolescents with ASD less likely to make social attributions, especially for those animations with the most complex social interactions. When stimuli were accompanied by music, both groups were equally impaired in appropriateness and intentionality.	Δ longer description time for animation with complex social interactions in ASD: *p* < 0.01; Δ appropriateness: *p* = 0.14; Δ intentionality: none reported.
3	Boorom et al. (2020), USA [56]	Parent-child dyads (*n* = 12)	24	Experimental design	ADOS-2, MSEL, video recordings, Child attentional leads and parent responsiveness, musical engagement	No significant difference between overallparental responsiveness in musical conditioncompared to non-musical condition. Parents provided significantly more physical play responses and significantly fewer verbal responses during musical vs. non-musical engagement with their child.	∆ overall parental responsiveness: *p* = 0.16; ∆ physical responses: *p* = 0.02; ∆ verbal responses: *p* < 0.001
4	Brown (2017), USA [57]	Children with ASD (*n* = 20); TD children (*n* = 30)	50	Experimental design	Face photographs, video recordings of children, a stringent procedure for 4 music stimuli	Across both conditions, TD children rated the happy faces as happier and the sad faces as sadder than children with ASD. Children with ASD took longer to respond when listening to sad music	Happy/sad ratings: *p* < 0.03
5	Buday (1995), USA [58]	Children with ASD (*n* = 10)	10	Experimental design	Number of signs imitated and spoken words (music or rhythm condition)	Music had a positive effect on the number of signs subjects were able to correctly imitate. The same positive effect was found for music in terms of the number of spoken words correctly imitated.	Significant main effect for condition type (music or rhythm)
6	Heaton (2003), United Kingdom [59]	Experiment 1 and 2: Children with ASD (*n* = 14); experiment 3: Children with ASD (*n* = 15)	29	Experimental design	PPVT, Raven’s Matrices, verbal IQ, stimuli and pitch identification task	Experiment 1: enhanced pitch memory and labelling in the ASD group; Experiment 2: subjects pre-exposed to labelled individual tones: superior chord segmentation; Experiment 3: When performance was less reliant on pitch memory, no group differences emerged.	Results not significant.
7	Katagiri (2009), USA [22]	Students with ASD (*n* = 12)	12	Cross-sectional design	Pre and post test including 4 sub-tests	All students with ASD improved significantlyin their understanding of the four selectedemotions. Background music was significantlymore effective than the other three conditionsimproving participants’ emotional understanding.	Δ understanding of the four selected emotions: *p* = 0.01; effect of background music: *p* = 0.01
8	Lundqvist et al. (2009), Sweden [60]	Adults with developmental disabilities (*n* = 20); of these with ASD (*n* = 10)	20	Randomised controlled trial	BPI, vibroacoustic treatment, assistant rating form	Vibroacoustic music reduced challenging behaviour in individuals with ASD and developmental disability demonstrated in BPI ratings, behaviour observation analyses, and assistants’ ratings. Vibroacoustic music was also shown to reduce the frequency of self-injurious behaviour (SIB).	Δ BPI SIB frequency: *p* < 0.05
9	Portnova et al. (2018), Russia [61]	Children with ASD (*n* = 21) and TD peers (*n* = 21)	42	Experimental Design	CARS, WISC-IV, EEG assessments, Six musical fragments, emotional self-report scale	Children with ASD assessed most music fragments similarly to their TD peers, with likelihood of EEG oscillatory patterns closely corresponding to emotion self-reports. In S2 fragments, "sad" was a reported emotion in TD children and adult neurotypical raters, but “angry and frightening” were emotions elicited by children with ASD. In S2 fragments, EEG oscillatory response showed greater cortical activation in the right hemisphere.	Emotional response to music self-report data: *p* < 0.00001
10	Stephenson et al. (2016), USA [62]	Students with ASD (*n* = 50) Adolescents with ASD (*n* = 41)	91	Experimental design	ADOS, WASI, SRS, SSP, music-evoked emotion recognition task, SAM, short task for baseline skin conductance level, SCR recording, SCAS-P	Participants with ASD showed a decrease in skin conductance response to music-evoked condition. Younger groups, regardless of diagnosis, showed greater physiological reactivity to scary stimuli than other emotions. A significant interaction of age group and diagnostic group was found.	Δ reaction to scary stimuli: *p* = 0.01. Δ interaction of age and diagnosis: *p* = 0.04
11	Thompson & Abel (2018), Australia [63]	Children with ASD (*n* = 16)	16	Experimental design	Diagnostic assessments, song and story recordings, gaze recording	Based on dwell time and fixation counts, children looked significantly at the performer’s face and body and less at the prop during singing than storytelling and when there was familiar material than unfamiliar material.	Δ dwell time: *p* = 0.02 in face and body looking in singing; Δ familiarity in singing: *p* = 0.02 Δ interaction between familiarity and body face looking: *p* = 0.020; Δ familiarity and looking: *p* = 0.005
12	Wagener et al. (2021), Luxembourg [64]	Children with ASD (*n* = 19); TD children (*n* = 31)	50	Experimental design	AQ-10 Child, 10 novel items based of emotion reactivity scale, facial stimuli from Pictures of Facial Affect, 2 music pieces, facial recognition task	Children with ASD had higher reaction times than controls; accuracy differed when incongruent or no music was played.	Significantly negative correlation between AQ-10 and emotion recognition accuracy.
13	Weiss et al. (2021), Canada [65]	Children with ASD (*n* = 26); TD Children (*n* = 26); adolescent or adult with WS (*n* = 26)	52	Experimental design	Western folk melodies, two musical games, unexpected recognition test	Both groups significantly distinguished the old melodies from the new melodies; they differed in overall memory. Children with ASD showed enhanced processing of socially significant auditory signals in the context of music.	∆ overall memory: *p* = 0.026; ∆ processing of auditory signals: *p* = 0.014
14	Woodman et al. (2018), USA [66]	Students with ASD (*n* = 13)	13	Pilot study on effects of exercising; experimental design	SIB-R, W-ADL, AQ-Child, Metabolic Equivalent of Tasks	Exercise intensity was highest during thestructured exercise periods and during slowmusic condition	Δ exercise intensity: *p* = 0.02
	**(3) Effect of MT and Musical Training on People with ASD**						
1	Allen et al. (2012), Australia [67]	Adults with ASD (*n* = 23); TD Adults (*n* = 24)	47	Experimental design (including 2 experiments)	BVAQ-B, AQ, 12 music items	Adults with alexithymia and ASD did not differ based on physiological responsiveness; ASD group significantly lower on the verbal measure; significant effect of mood with music in response between two groups but the maximum difference was relieved for scary music.	Δ verbal measure: *p* < 0.05
2	Bettison (1996), Australia [68]	Children with ASD (*n* = 80)	80	Longitudinalstudy design	ABC, DBC, PPVT, SSQ, LIPS, SP, SD, concomitant variables	Children in auditory training showed a significant improvement in all measurements after 1 month. They showed a significant improvement in ABC and DBC (teacher) between 3 and 6 months; this improvement was not observed after 12 months of interventions.	Δ after 1 month: none reported Δ ABC scores between the 3-and 6-month assessments: *p* < 0.003; Δ DBC (teacher) mean scores between the 3-and 6-month assessments: *p* < 0.04; Δ SSQ score at 1 month: *p* = 0.05
3	Bharathi et al. (2019), India [69]	Children with ASD (*n* = 54)	54	Experimental design	CARS, TSSA, active MT intervention, passive MT intervention,	Significant improvement in the social skills of the active MT group during the post-test phase. It was significantly greater than the children of the passive MT group.	Increase in TSSA social skills scores (*p* < 0.05)
4	Bieleninik et al. (2017), Norway [70]	Children with ASD (*n* = 364)	364	Randomised controlled trial	TIME-A, ADOS, IQ formal test, SRS, QoL	Children with ASD in MT did not result in significant improvement in mean symptom scores compared to enhanced standard care.	Δ improvement in MT: *p* = 0.88
5	Boso et al. (2007), Italy [71]	Young adults with ASD (*n* = 8)	8	Longitudinalstudy design	CARS, CGI, BPRS, 5-point Liker-type scale for musical skills	After 52 weeks of training, significant improvement in both the CGI and BPRS scales as well as patient’s musical skills compared to baseline ratings.	Δ CGI: *p* < 0.05; Δ BPRS: *p* < 0.05; Δ musical skills: *p* < 0.05
6	Carpente (2017), USA [72]	Children with ASD (*n* = 4)	4	Experimental design	FEAS pre and post-test, 26 DIR based IMT sessions	Improvements in areas of self-regulation, engagement, behavioural organization, and two-way purposeful communication	None reported.
7	Cibrian et al. (2020), USA [73]	Children with ASD (*n* = 22)	22	Pilot randomised controlled trial	DCDQ, PiT, timing, synchronisation assessment, the strength control assessment	Significant improvement in coordination with greater control of their movements.	∆ coordination: *p* = 0.003
8	Cook et al. (2019), United Kingdom [74]	Children with ASD (*n* = 55) TD children (*n* = 10)	65	Cross sectional study design	Demographics, SBQ, Child-Report Sympathy Scale, the victim scale, the bully scale, vignette reading, Open-format questions	In response to hypothetical scenario depicting social exclusion of child with ASD, TD children in the contact group showed a greater increase in prosocial emotions; a greater decrease in tendency to be a victim than those in the no-contact group with 19.7% reduction in victimisation.	Δ prosocial behaviour: *p* = 0.04; Δ tendency to be victim: *p* = 0.01; Δ victimisation: *p* = 0.07
9	Crawford et al. (2017), United Kingdom [75]	Children with ASD (*n* = 364)	5	Randomised controlled trial	Enhanced standard care, ADOS social affect, low frequency IMT, high frequency IMT	From baseline to 5 months, mean scores of ADOS social affect decreased from 14.1 to 13.3 in music therapy and from 13.5 to 12.4 in standard care.	Δ improvement in ADOS social affect: no significant difference; Δ parent-rated social responsiveness score: no significant difference
10	Edgerton (1994), USA [76]	Children with ASD (*n* = 11)	11	Reversal design	CRASS, IMT	Strong efficacy of IMT in increasing autistic children’s communicative behaviours.	Δ CRASS scores: *p* < 0.01
11	Finnigan & Star (2010), Canada [26]	Child with ASD (*n* = 1)	1	Alternating Treatment Design	MSEL, CARS, ADOS, VABS-II, MT intervention	MT was more effective than no MT in increasing all three social responsive behaviours; no avoidant behaviours were observed during the music condition.	None reported.
12	Gattino et al. (2011), Brazil [77]	Children with ASD (*n* = 24)	24	Randomised controlled trial	ADI-R, CPM, Music therapy assessment sessions, Relational Music therapy, CARS, CARS-BR	Effects of relational music therapy on communication skills of ASD children inconclusive.	n.s
13	Ghasemtabar et al. (2015), Iran [78]	Children with ASD (*n* = 27)	27	Longitudinal study design	CARS, SSRS-P for elementary period	In pre-post-test, significant increase in social skills scores.	Δ increase in social skills: *p* < 0.001
14	Jin et al. (2020), China [79]	Children with ASD (*n* = 60)	60	Randomised controlled trial	ATEC, ABC	After intervention, the scores of each item of the social domain in ATEC and the scores of ABC; feeling, communication, physical movement, language, and healthy behaviour were lower than those before intervention in both groups.	∆ ATEC and ABC scores: *p* < 0.01; ∆ before intervention in both groups: *p* < 0.05
15	Kalas (2012), USA [80]	Children with ASD (*n* = 30)	30	Cross-sectional design	ESCS, RJA	Significant interaction between music modality and functioning level was found indicating that the effect of simple versus complex music was dependent on functioning level.	Δ severe ASD in RJA score in simple music condition: *p* = 0.004; Δ mild to moderate RJA score in complex music condition: *p* = 0.011
16	Kern & Aldridge (2006), Canada [81]	Children with ASD (*n* = 4)	4	Experimental design	CARS, musical adaptation of a child care playground, individually designed MT intervention	Musical adaptation of the playground did not improve social interactions of ASD children, but it facilitated their play and involvement with peers by attraction to the sound and opportunity to use the instruments. The song interventions produced desirable peer interaction outcomes.	None reported.
17	Kim et al. (2009), South Korea [28]	Children with (*n* = 10)	10	Randomised controlled trial	CA, Korean version of the CARS, DQs, PEP, SQs, SMS, DVD recording, TV monitor	MT improved social, emotional and motivational development in children with ASD.	Δ joy behaviour: *p* < 0.001; Δ effect of emotional synchronicity: *p* < 0.001; Δ effect of initiation of engagement behaviour: *p* < 0.001; Δ effect of initiation of interaction by the therapist: *p* < 0.001; Δ compliant response: *p* < 0.001; Δ no response: *p* < 0.001
18	Kim et al. (2008), South Korea [82]	Children with ASD (*n* = 13)	13	Randomised controlled trial	Korean version of the CARS, ADOS, PDDBI, ESCS	IMT was more effective at facilitating joint attention behaviours and non-verbal social communication skills in children than playing. Significantly more and lengthier events of eye contact and turn-taking in IMT than play sessions.	Δ efficacy of IMT on joint attention behaviours: *p* = 0.01 Δ eye contact: *p* < 0.0001; Δ turn-taking: *p* < 0.0001
19	LaGasse (2014), USA [15]	Children with ASD (*n* = 17)	17	Randomised controlled trial	CARS2, SRS, ATEC, video recordings for eye gaze, joint attention and communication	Significant between-group differences in joint attention with peers and eye game towards people, with significant improvement in participants in the MTG; no significant difference for communication, response to communication or social withdraw/behaviours.	Δ SRS score: *p* = 0.018, Δ eye gaze: *p* = 0.022; Δ joint attention: *p* = 0.031; Δ initiation of communication with another peer: *p* > 0.05; Δ initiation with an adult: *p* > 0.05; Δ response to communication: *p* > 0.05; Δ social withdraw/behaviours: *p* > 0.05
20	LaGasse et al. (2019), USA [83]	Children with ASD (*n* = 7) TD children (*n* = 7)	14	Experimental study design	EEG recording with sensory gating paradigm, EEG data acquisition, TEA-Ch, MTA protocol	The initial outcomes of brain responses and behaviours showed a positive effect of MT on selective attention skills.	Δ TEA-Ch scores in post-intervention: *p* = 0.025; Δ positive correlation between P50 difference and attention scores in post-intervention: *p* = 0.042
21	Lense et al. (2020), USA [84]	Children with ASD (*n* = 14) TD children (*n* = 14)	28	Mixed design with survey and experimental design	ADOS-2, Serenade program, 14-item program evaluation survey, semi-structured interview, video recordings	Parent-child music interventions improved family well-being with enhancing the value of integrated community participation experiences at the level of the family structure.	N/A
22	Lim (2010), USA [85]	Children with ASD (*n* = 50)	50	Randomised controlled trial	CARS, ADI-R, Preschool language scale, EROWPV, fill-in-the-blank task, pictures from PECS, DSLM, VPES	In both conditions, participants improved their pre to post-test verbal production. No significant difference between low and high functioning in improving speech production after both training.	Δ improvement in pre to post-test production: *p* < 0.001; Δ n.s difference between low and high functioning in speech production: *p* = 0.053
23	Lim & Draper (2011), USA [86]	Children with ASD (*n* = 22);	22	Cross-sectional design	VPES	Both music and speech training were effective for production of the four ABA verbal operants; difference between music and speech training not significant. Music incorporated ABA VB training.	Δ echoic production: *p* = 0.039
24	Mendelson et al. (2016), USA [87]	Children with ASD (*n* = 5) Children with intellectual disability (*n* = 32)	37	Cross-sectional design	Voices together program, behavioural observations, SSIS-RS	Both groups showed increases in verbal responses over time; only long-term group showed significant within-group increases.	Δ within-group increases: *p* < 0.05
25	Moradi et al. (2018), Iran [88]	Children with ASD (*n* = 48)	48	Randomised controlled trial	Elecsys method, research kits of the human NGF, RCPM	Significant improvement in NGF level in participants of music condition, but not significantly different to the placebo group. The level of NGF in participants in vitamin D condition showed a significant increase compared to participants in music condition. The level of NGF in music and vitamin D condition was significantly higher compared to only music, only vitamin D and placebo condition.	Δ improvement in NGF level in music condition: *p* = 0.001; Δ no difference between music condition and placebo group: *p* = 0.07; Δ participants’ higher NGF level in vitamin D condition compared to music condition: *p* = 0.001; Δ NGF level in both music and vitamin D condition compared to only music only vitamin D or placebo group: *p* = 0.001.
26	Mössler et al. (2019), Norway [89]	Children with ASD (*n* = 48)	48	Observational longitudinal design	ADOS, ADI-R, AQR, ADOS-G, SRS	Significant time dependent main effect of AQR in the ADOS social effect was found after 5 months. This interaction was also significant after 12 months. There was an improvement in SRS total over the course of therapy when there was a higher AQR match rate	∆ social effect after 5 months: *p* = 0.0162; after 12 months: *p* = 0.0399; ∆ SRS: *p* = 0.0426
27	Pedregal & Heaton (2021), United Kingdom [90]	Children with ASD (*n* = 11)	11	Pilot randomised controlled trial	BPVS-III, EAQ, ER, music intervention (group)	Chronological age (CA) and receptive vocabulary were significantly associated with recognition of facial and verbal emotions and not hiding emotions. At post-test, older children showed a greater increase in recognition of voices and in emotional bodily awareness.	∆ Vocal and facial ER scores: *p* < 0.01; ∆ Not hiding emotions score: *p* < 0.01
28	Poquérusse et al. (2018), Singapore [91]	Adults with ASD (*n* = 20)	20	Experimental design (2 studies were conducted; MT was used in the second study)	sAA	Occupational therapy leaded a significant increase in sAA levels while MT significantly deceased baseline sAA levels indicating that the ability of receding stress in both interventions and by proxy contribute to improve overall well-being.	Δ sAA levels in occupational therapy: *p* < 0.05; Δ sAA levels in MT: *p* < 0.05
29	Rabeyron et al. (2020), France [92]	Children with ASD (*n* = 36)	36	Randomised controlled trial	CGI, CARS, ABC, CGI-I	Significant decrease in CGI scores which was more pronounced in MT group than ML group; a significant decrease found for CARS and ABC scores in both groups with lack of significant interaction between group and time respectively.	∆ CGI scores: *p* < 0.001; ∆ effect of CGI between MT group and ML group: *p* = 0.017; ∆ CARS and ABC scores for both groups: *p* = 0.001; ∆ no significant interaction between group and time: *p* = 0.65 and *p* = 0.85 respectively
30	Rosenblatt et al. (2011), USA [93]	Children with ASD (*n* = 24);	24	Cross-sectional design	BASC-2, ABC	The post-treatment scores on the atypicality score of the BASC-2, which measures some of the core features of autism, changed significantly.	Δ core features of autism: *p* = 0.003
31	Ruan et al. (2018), China [94]	Children (*n* = 40,273)	40.273	Survey	Self-administered structured questionnaire, ABC, covariates	Antenatal music training and maternal talk showed significant reduction in autistic-like behaviours in children.	Δ often antenatal MT and ABC correlation: *p* = 0.003; occasionally: *p* = 0.993; Δ often maternal talk and ABC correlation: *p* < 0.001; occasionally: *p* = 0.0002
32	Schwartzberg & Silverman (2016), USA [95]	Children with ASD (*n* = 29)	29	Randomized control trial	Short story intervention, singing intervention, CC scores	Mean CC scores increased from day one to day 3 for both the control and experimental groups. Baseline CC on day 1 was enhanced by music but musically induced gains declined for days 2 and 3, possibly due to learning effects and repetition of material over the course of three days.	Δ CC scores: *p* < 0.05
33	Sharda et al. (2018), Canada [21]	Children with ASD (*n* = 51)	51	Longitudinal study design	ADOS, ADI-R, CARS, Clinical assessment, SRS-II, CCC-2,VABS-MB, FQoL, WASI-II, IQ, CLEF-4, PPVT-4, Montreal Battery for Evaluation of Musical Abilities, MRI scan	8-12 weeks of individual music intervention improved social communication and functional brain connectivity.	Δ communication score: *p* = 0.01; Δ subcortical activity: *p* < 0.001; Δ auditory and frontal-motor regions activity: *p* < 0.0001
34	Simpson et al. (2013), Australia [20]	Children with ASD (*n* = 22)	22	Randomised controlled trial	SCQ, SIB-R, EVT-2, PPVT-4, singing intervention, spoken intervention	Children with autism were more engaged in the sung condition compared to the spoken condition. The use of infant-directed singing was more engaging for children with ASD over infant-directed speech.	Δ engagement levels between groups: *p* = 0.04
35	Simpson et al. (2015), Australia [96]	Children with ASD (*n* = 22)	22	Crossover study design	SCQ, EVT2, PPVT-4, SIBR, infant-directed song intervention, infant-directed speech intervention	There was no significant difference between the sung and spoken conditions. There was a significant increase in receptive labelling skills after both conditions were enacted and these results were maintained at follow-up. A difference in group performance was found.	n. s.
36	Srinivasan et al. (2015), USA [97]	Children with ASD (*n* = 36)	36	Randomised controlled trial	SCQ, ADOS-2, VABS, RBS-R, ABA, PECS, TEACCH	With training, the rhythm group showed a reduction in negative affect and an increase in interest affect and positive affect.	Δ negative affect: *p* < 0.05; Δ interest affect: *p* < 0.002; Δ positive affect: *p* < 0.002
37	Thompson et al. (2014), Australia [98]	Children with ASD (*n* = 23)	23	Randomised controlled trial	VSEEC, SRS-PS, MBCDI-W&G, PCRI, MTDA, semi-structure interview	FCMT improves social interactions in home, the community and the parent-relationship whereas no improvement in language skills or general social responsiveness.	Δ VSEEC score: *p* = 0.001; Δ interpersonal improvement: *p* = 0.001; Δ SRS-PS: n.s; Δ language skills: n.s; Δ parent-child relationship: n. s.
38	Venuti et al. (2017), Italy [99]	Children with ASD (*n* = 25)	25	Experimental design	DSM-IV-TR assessment, the ADOS, and the Griffiths Mental Development Scales, 20 IMT sessions, observational tool for coding behaviours (CBEC and ABEC)	There was an increase in the amount of synchronic activity throughout the IMT sessions, with a significant difference from Session 1 to Session 20 in behavioural synchrony and emotional attunement.	ΔCm at T1 and at T2: n.s. ΔCm at T1 and T3: *p* < 0.01
39(a)	Yoo & Kim (2018), South Korea (This publication reports two different studies) [100]	Children with ASD (*n* = 10) TD children (*n* = 42)	52	Experimental design	PRI, K-WISC-IV, K-CARS, K-SSRS, KDEF, drum tapping tasks	The presence of rhythmic cueing and tempo adjustment correlated with social skills, providing a strong rationale for the use of dyadic drum playing to address social skills.	Δ cooperation: *p* = 0.046; Δ self-control: *p* = 0.028; Δ total score of K-SSRS: *p* = 0.028
39(b)		Children with ASD (*n* = 8)	8	Experimental design	K-CARS, asynchrony measures during tapping tasks	Children shows a reduction in asynchrony when tapping with a partner at adjusted tempi after the rhythm-mediated intervention and showed a greater engagement in joint action following the intervention.	None reported.
	**(4) Combination of MT and Dance Movement Therapy in Patients with ASD**						
1	Bergmann et al. (2021), Germany [101]	Adults with ASD (*n* = 12)	12	Experimental design	Pre and post self-assessment, DAS, CSQ-8, novel AutCom intervention and questionnaire, SRS, ABC, MOAS, POS	No patients withdrew from the AutCom training group and the client satisfaction questionnaire resulted in a mean score of 30 out of 32 points. Significant improvement in social competence compared to the control group and emotional competence in the pre-post self-assessment on the AutCom questionnaire. No significant improvement in challenging behaviour and quality of life.	Significant group differences in AutCom questionnaire: *p* = 0.024
2	Lakes et al. (2019), USA [102]	Children with ASD (*n* = 12)	12	Longitudinal study design	RCS, observer-rated measure of child self-regulation, RBS-R, PACES, video recording, post intervention PACES	Group-level reductions in stereotyped and compulsive behaviours of 8% and 4%, respectively; post-hoc analysis showed substantial individual differences in children’s responses to the intervention.	Δ compulsive behaviours: *p* = 0.02; Δ stereotyped behaviours: *p* = 0.02
3	Mateos-Moreno & Atencia-Dona (2013), Spain [103]	Children with ASD (*n* = 16); TD children (*n* = 8)	24	Experimental design	CARS, 36 sessions of combined MT and DMT, ECA-R	Positive trend towards a reduction in disorder scores in both control and experimental groups; all participants were attending their regular therapies and receiving pharmacologic treatments during the experimental period.	Δ ECA-R: *p* < 0.05

ABA: Applied Behaviour Analysis; ABA VB: Applied Behaviour Analysis Verbal Behaviour; ABC: Aberrant Behaviour Checklist; ABEC: Adult Behavioural and Emotional status Code; ADI-R: Autism Diagnostic Interview-Revised; ADOS: Autism Diagnostic Observation Schedule; ADOS-G: Autism Diagnostic Observation Schedule-Generic; ADOS-R: Au-tism Diagnostic Observation Schedule–Revised; ADOS-2: Autism Diagnostic Observation Schedule, Second Edition; ALI: autism and co-morbid language impairment; AQ: Autism Quotient; AQ-Child: Autism Spectrum Quotient: Children’s Version AQR: Assessment of the Quality of Relationship; AQ-10: Autism Quotient-10 items; ASD: Autism Spectrum Disorder; ATEC: Autism Treatment Evaluation Checklist; AutCom: Autism-Competence-Group; BAIS: Bucknell Auditory Imagery Scale; BAIS-C: Bucknell Auditory Imagery Scale Control; BAIS-V: Bucknell Auditory Imagery Scale Vivid; BASC-2: Behavioural Assessment System for Children-Second Edition; BPI: Behavioural Problems Inventory; BPRS: Brief Psychiatric Rating Scale; BPVS: British Picture Vocabulary Scale; BPVS-II: British Picture Vocabulary Scale: Second Edition; BVAQ-B: The Bermond-Vorst Alexithymia Question-naire - version B; CBEC: Child Behavioural and Emotional status Code; CC: Comprehension Checks; CCC-2: Children communication Checklist; CGI: Clinical Global Impression; CGI-I: Clinical Global Impression Improvement; CLEF-4: Clinical Evaluation of Language Fundamentals; CMS: Child Memory Scale; CPS: Closure Positive Shift; CRASS: Checklist of Communicative Responses/Acts Score Sheet; CSQ-8: Client Satisfaction Questionnaire; DAS: Disability assessment schedule; DBC: Developmental Behaviour Checklist; DCDQ: The Developmental Coordination Disorder Questionnaire; DIR: Developmental Individual Difference Relationship based model; DQs: Developmental Quotients; DMT: Dance Movement Therapy; DSLM: Developmental Speech and Language Training through Music; DVD: Digital Versatile Disc; EAQ: Emotion awareness questionnaire; ECA-R: Evalua-tion of Autistic Behaviour; EEG: electroencephalography; ER: Emotion recognition test; EROWPV: Expressive and Receptive One Word Picture Vocabulary Test; ERP: early recep-tor potential ESCS: Early Social Communication Scales; EVT: Expressive Vocabulary Test-standard score; FEAS: Functional Emotional Assessment Scale; FCMT: Family-centred music therapy; FQoL: Beach Family Quality of Life Scale fMRI: functional magnetic resonance imagining; HFA: high functioning autism; GADS: Asperger Gilliam Asperger’s Dis-order Scale; IGF: inferior frontal gyrus; IQ: Intelligence Quotient; IMT: Improvisational music therapy; KADI: Krug Asperger's Disorder Index; K-CARS: Korean-Childhood Autism Rating Scale; KDEF: Karolinska Directed Emotional Faces; K-SSRS: Korean-Social Skills Rating System; K-WISC-IV: Korean Weschler Intelligence Scale for Children-IV; LPS: Per-formance Testing System; LIPS: Leiter International Performance Scale; MBCDI-W&G: The MacArthur-Bates Communicative Development interventions, Words and Gestures; ML: Music listening group; MOS: Modified overt aggression scale; MSEL: Mullen sales of early learning; MT: music therapy; MTA: Music Therapy Attention; MTDA: Music Therapy Diagnostic Assessment; MTG: Music therapy group; MRI: magnetic resonance imagining; MWB: Multiple Choice Vocabulary Test-B; *n*: Number of participants; N: Total number of study participants; N/A: Not available; NGF: Beta-nerve growth factor; n.s: Not significant; *p*: *p*-value; PACES: Physical Activity Enjoyment Scale; PCRI: Parent-Child Relationship Inventory; PECS: Picture Exchange Communication System; PDDBI: Pervasive Developmental Disorder Behaviour Inventory-C; PEP: Psychoeducational Profile; PiT: Personality Item Test; PEMM: Perception of Emotions and Movement in Music; PEPS: Profiling Elements of the Prosodic System; POS: Personal Outcome Scale; PPVT: Peabody Picture Vocabulary; PPVT-III: Peabody Picture Vocabulary Test - Third Edition; PPVT-4: Peabody Picture Vocabulary Test-Fourth Edition; PRI: A Perceptual Reasoning Index; QoL: Quality of Life; RBS-R: Repetitive Behaviour Scale-Revised; RCS: Response to Challenge Scale; RJA: Responding to Joint Attention; RSPM: Raven’s Standard Progressive Ma-trices; sAA: Salivary α-amylase; SAM: Self-Assessment Manikin; SAMMI: Salk and McGill Music Inventory SBI-R: Scales of Independent Behaviour Revised; SBQ: Social Behaviour Questionnaire; SCQ: Social Communication Questionnaire; SD: sound distress; SLI: specific language impairment; SIBR: Scales of Independent Behaviour Early Development Form; SMS Social Maturity Scale; SP: Sensory Problems Checklist; SQs: Social Quotients; SRS: Social Responsiveness Scale; SRS-PS: The Social Responsiveness Scale-Preschool Ver-sion for 2-years-olds; SSIS-RS: Social Skills Improvement System-Rating Scale; SSQ: Sound Sensitivity Questionnaire; SSRS-P: Social Skills Ration System Scale for elementary peri-od; STAI: State-Trait Anxiety Inventory; TD: typically developed; TD-NH: Typical-development and normal hearing; TEA-Ch: Test of Everyday Attention for Children; TEACCH: Treatment and Education of Autistic and Related Communication-Handicapped Children; TIME-A: The Trial of Improvisational Music Therapy With Autism; TV: television; VABS: Vineland Adaptive Behaviour Scale; VIQ: Verbal IQ; VPES: Verbal Production Evaluation Scale; VSEEC: Vineland Social-Emotional Early Childhood Scales; W-ADL: Waisman Activities of Daily Living Scale; W-ADL-R: Waisman Activities of Daily Living Scale-Revised; WAIS-III: Wechsler’s Adult Intelligence Scale-Third Edition; WASI: Weschler Abbreviated Scale of Intelligence; WASI-II: Wechsler’s Abbreviated Intelligence Scale-Second Edition; WASI-R: Wechsler’s Abbreviated Intelligence Scale–Revised; WISC-IV: Wechsler Intelligence Scale for Children-Fourth Edition.

## Data Availability

Data availability is not applicable to this article because no new data were created or analysed in this study.

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
