# Peer review of "A Systematic Review of Scientific Studies on the Effects of Music in People with or at Risk for Autism Spectrum Disorder"

_ijerph, 2022, doi:10.3390/ijerph19095150_

Round 1

Reviewer 1 Report

The content of the article provides a valuable review of the literature on the topic. The article was edited correctly. The text of the article is very large, it is worth re-examining it to shorten it. The final conclusions seem to be too general.

The analysis of the material took place in terms of three

 areas:

  1. Music Perception in ASD
  2. Effects of music in people with ASD
  3. Effect of MT and musical training on people with ASD

In the conclusion of the article, the authors divide the process of inference into: discussion, limitations and conclusions. There was no analysis of the conclusions in terms of the three areas analyzed in the text of the article. Some outline of this analysis was revealed in the discussion section, but in my opinion it was treated too generally. I suggest detailing the conclusions by aspect of the three analyzed areas in the text.

Author Response

Reviewer: 1
The content of the article provides a valuable review of the literature on the topic. The article was edited correctly. The text of the article is very large, it is worth re-examining it to shorten it. The final conclusions seem to be too general.

The analysis of the material took place in terms of three areas:

  1. Music Perception in ASD
  2. Effects of music in people with ASD
  3. Effect of MT and musical training on people with ASD

In the conclusion of the article, the authors divide the process of inference into: discussion, limitations and conclusions. There was no analysis of the conclusions in terms of the three areas analyzed in the text of the article. Some outline of this analysis was revealed in the discussion section, but in my opinion it was treated too generally. I suggest detailing the conclusions by aspect of the three analyzed areas in the text.

Response to reviewer 1:

  • We agree with the reviewer, and we noticed that we made a mistake on page 20, when we wrote we had identified three categories. In fact, we found 4 categories:
    1. Music perception in ASD
    2. Effects of music in people with ASD
    3. Effects of MT and musical training on people with ASD
    4. Combination of MT and Dance Movement Therapy in patients with ASD
  • In accordance with this structure of the findings, we have now re-organized the results section: Paragraph 3.1. explains the included studies, paragraph 3.2. summarises methodological aspects of included studies, and paragraph 3.3. sums up the content of the studies. This paragraph 3.3. of the revised version has now for sub-sections:

3.3.1. Music Perception in ASD

3.3.2. Effect of Music on People with ASD

3.3.3. Effects of MT and Musical Training on People with ASD

3.3.4. Effects of combined MT and Dance Movement Therapy

  • The discussion and the conclusion are now following the same pattern. In the discussion we cover these four categories in section 4.1. The conclusion is a paragraph separate from the discussion in accordance with the manuscript guidelines of the journal.

We thank reviewer 1 for their comments which have helped to structure the manuscript in a more understandable and logical way.

Reviewer 2 Report

This is an important and badly needed systematic review in the field of MT and ASD. And in spite of the tremendous heterogeneity in the field, I think the authors did a serious and effective work. But this project could be even more interesting and attracting different health professional and caregivers, with some modifications in the format of presentation and adding more clear information ,that are already mentioned en passant in the text:

In the introduction : the information from line 37 to 50 should be clarify and perhaps a synthesis of the new findings are necessary: perception and expression of music is definitively a brain changing effect.

From line 87 to 96, the info should be more to the point .Looks vague to me.pe: which activation areas? where in the cortex? in cerebellum ?.

I would definitively suggest a change in the format of the presentation of the well investigated studies:should be presented in chronological and not in alphabetical order: not only are distracting the objectives but also are not focusing in the recent advances in the field.

The authors should try to look into the data for possible differences in the adolescents and adults versus the children with ASD and MT.

Author Response

Reviewer 2: This is an important and badly needed systematic review in the field of MT and ASD. And in spite of the tremendous heterogeneity in the field, I think the authors did a serious and effective work. But this project could be even more interesting and attracting different health professional and caregivers, with some modifications in the format of presentation and adding more clear information, that are already mentioned en passant in the text:

In the introduction: the information from line 37 to 50 should be clarified and perhaps a synthesis of the new findings is necessary: perception and expression of music is definitively a brain changing effect.

Response:

  • Thank you for your comment. For added clarity, lines 37-50 now read, “Over the past 50 years, epidemiological studies have confirmed the prevalence of ASD is increasing globally. The current prevalence of ASD reported by the Center for Disease Control is estimated to be about one in 160 children. ASD is more than four times more prevalent among males than females [4].
  • According to the Diagnostic and Statistical Manual of Mental Disorders (DSM-5), ASD is characterised by a persistent deficit in social communication, restrictive and repetitive patterns of behaviours, social and occupational impairments, and other areas of functioning [5].
  • While the exact cause for ASD remains undetermined, literature on developmental disorders indicates that brain abnormalities could explain the reason for ASD in structure, function, and genetic susceptibility [6]. The literature further suggests that ASD is genetically determined. Additionally, environmental influences might also be considered as risk factors for developing ASD [7] and there is an interaction effect of genes and environmental components [8].”

Reviewer 2: From line 87 to 96, the info should be more to the point. Looks vague to me.pe: which activation areas? where in the cortex? in cerebellum?

Response:

  • We agree that this paragraph can come across vague when speaking about the brain regions activated by active and receptive music making. The following sentence has been added for clarification, “In both receptive and active music making experiences, there is an activation in the superior temporal lobe and inferior frontal areas of the brain involved in cognitive, sensorimotor, and perception-action mediation areas by increasing the synchrony between these cortical areas, which in turn, promote heightened sensory integration [19].”

Reviewer 2: I would definitively suggest a change in the format of the presentation of the well investigated studies: should be presented in chronological and not in alphabetical order: not only are distracting the objectives but also are not focusing in the recent advances in the field.

Response:

  • We thank the reviewer for their comment. However, we disagree with this suggestion for two reasons:

1) We have followed the PRISMA statement (Reference #30: Moher et al: Preferred Reporting Items for Systematic Reviews and Meta-Analyses: The PRISMA Statement. PLoS Medicine, 2009b, 6(7), e1000097). The practicalities of this statement were further explained in Liberati et al. The PRISMA statement for reporting systematic reviews and meta-analyses of studies that evaluate health care interventions: Explanation and elaboration. PLoS Med. 2009;6:e1000100. Table 2 of this article gives an example how the summary of study characteristics should be presented, where the articles are presented in alphabetical order.

2) We have also checked other systematic reviews, and almost all of them had listed the included articles in alphabetical order.

It is easier when searching for the articles cited in the text to search for the name of the first author for various reasons. For example, in some years, several articles have been published on the topic. 

Therefore, we chose to present the articles in alphabetical order in order to follow the guidelines and for practical reasons.

We agree with the reviewer that it would be easier to find the latest study if the papers were presented in chronological order. But, unfortunately, we cannot do it both ways.

Reviewer 2: The authors should try to look into the data for possible differences in the adolescents and adults versus the children with ASD and MT.

Response:

  • This is an important point. This paragraph has been added to the discussion section to account for the similarities and unaccounted for differences of both age groups, “Only 10 out of the 81 studies included focused on adults and the effects of music, MT and creative arts therapies on ASD [35,42,48, 50, 54, 55. 63, 66, 89, 101]. However, similar results were observed in comparison to adolescent ASD studies. For example, adults with ASD had a reduction in challenging behaviours associated with ASD [66] and significant improvements in social competence compared to control groups [101]. However, there is a clear distinction in researchers focusing on children and adolescents more than adults in music ASD studies. Future research should include more samples of adults and even studies involving both children and adults with ASD so accurate comparisons can be made about the effects of music and MT, their similarities, as well as their differences for these populations in different age categories.”

We thank reviewer 2 for the helpful comments, and we think that the manuscript has improved significantly as a result of the added information in response to their comments.